# Preoperative High Visceral Fat Increases Severe Complications but Improves Long-Term Prognosis after Gastrectomy for Patients with Advanced Gastric Cancer: A Propensity Score Matching Analysis

**DOI:** 10.3390/nu14204236

**Published:** 2022-10-11

**Authors:** Ryota Matsui, Noriyuki Inaki, Toshikatsu Tsuji, Yoji Kokura, Ryo Momosaki

**Affiliations:** 1Department of Gastroenterological Surgery, Ishikawa Prefectural Central Hospital, 2-1 Kuratsuki-higashi, Kanazawa 920-8530, Ishikawa, Japan; 2Department of Gastrointestinal Surgery/Breast Surgery, Graduate School of Medical Science, Kanazawa University, 13-1 Takara-machi, Kanazawa 920-8641, Ishikawa, Japan; 3Department of Nutritional Management, Keiju Hatogaoka Integrated Facility for Medical and Long-Term Care, Hosu 927-0023, Ishikawa, Japan; 4Department of Rehabilitation Medicine, Mie University Graduate School of Medicine, 2-174 Edobashi, Tsu 514-8507, Mie, Japan

**Keywords:** gastric cancer, obesity paradox, overall survival, postoperative complication, visceral adipose tissue

## Abstract

This study investigated the paradox of high visceral fat mass increasing severe complications but improving long-term prognosis after radical gastrectomy for gastric cancer. This was a retrospective cohort study of consecutive patients with primary stage I–III gastric cancer who underwent gastrectomy between April 2008 and June 2018. The visceral adipose tissue index (VAI) was calculated by dividing the visceral fat mass preoperatively measured on computed tomography by the square of the height. Patients with VAIs below the median cut-off value were classified as low-VAI, while those above it were classified as high-VAI. We compared the postoperative complication rate and overall survival (OS) in the low- and high-VAI groups after adjusting patient characteristics using propensity score matching (PSM). There were 155 patients in both groups after PSM. After matching, there was no significant difference in factors other than BMI and VAI that were not adjusted. The high-VAI group had more severe postoperative complications (*p* = 0.018), but the OS was significantly better in the high-VAI group (hazard ratio 0.611, 95%CI 0.403–0.928, *p* = 0.021). Preoperative high visceral fat mass not only increased severe complications, but also improved OS after gastrectomy in patients with advanced gastric cancer.

## 1. Introduction

In patients with gastric cancer, preoperative body composition has recently been shown to be a useful measure of postoperative complications following gastrectomy. When visceral fat content is high, infectious complications such as pancreatic fistula, intraabdominal abscess, anastomotic leakage, and wound infection increase [1,2,3,4,5,6,7,8,9,10]. The reason for this is that the greater the amount of visceral fat, the longer the operating time, the greater the amount of intraoperative blood loss, and the greater the difficulty of the operation [7,11,12]. Indeed, obese patients with c-stage I gastric cancer and metabolic syndrome who underwent gastrectomy after a one-month preoperative exercise program experienced a decrease in visceral fat mass and postoperative complications [13]. This suggests that having low visceral fat before surgery may help to prevent complications.

On the other hand, there is still no certain consensus on the impact of visceral fat content on long-term prognosis after gastrectomy in gastric cancer patients. According to reports, postoperative complications following a gastrectomy are linked to a poor long-term prognosis [14], and complications that are severe have a worse prognosis [15]. A high visceral fat content increases postoperative complications and may result in a poor long-term prognosis following gastrectomy. In cancer patients, obesity has been linked to the development of cancer [16], although paradoxical associations with prolonged long-term survival have also been observed [17]. Among cancers, the prognosis of lung and colorectal cancer is improved by obesity [18,19], while it has been linked to a poor prognosis in breast cancer, pancreatic cancer, and hepatocellular carcinoma [20,21,22]. Since obesity is a global issue, it is crucial to comprehend how visceral fat mass affects a patient’s long-term prognosis following gastrectomy for gastric cancer.

In this study, we investigated postoperative complications and overall survival (OS) in advanced gastric cancer according to preoperative visceral fat mass. We hypothesized that a high preoperative visceral fat content would increase postoperative complications and lead to poor OS in patients with advanced gastric cancer.

## 2. Materials and Methods

### 2.1. Study Design

This was a single-institution, retrospective cohort study conducted at Ishikawa Prefectural Central Hospital, which included consecutive patients who underwent gastrectomy for primary stage I–III gastric cancer, diagnosed according to the 15th edition of the Japanese Classification of Gastric Carcinoma, between April 2008 and June 2018. Clinical and laboratory data, including medical records and images, were collected retrospectively using the hospital’s electronic patient record system. As an inclusion criterion, we considered three factors: (1) primary gastric cancer; (2) gastrectomy; and (3) computed tomography (CT) images preoperatively. The following patients were excluded: (1) pathologically diagnosed early gastric cancer, (2) residual gastric cancer, (3) cancers of other organs, (4) preoperative treatment, (5) pStage IV, (6) non-gastrectomy surgical procedures, and (7) insufficient CT imaging data. The patients who met the criteria outlined above were divided into two groups based on their visceral fat levels. Using propensity score matching (PSM), we compared postoperative outcomes between the two groups after adjusting for patient background.

### 2.2. Postoperative Chemotherapy

The S-1 postoperative adjuvant chemotherapy regimen begun at 80–120 mg/m^2^ per day in p-stage II–III and reduced in accordance with recommendations if side effects were detected. S-1 postoperative chemotherapy was administered for a maximum of a year, and no additional therapy was provided until recurrence. In accordance with the Japanese Gastric Cancer Treatment Guidelines, chemotherapy was given to patients who experienced recurrences.

The outpatient clinic conducted patient follow-ups. Hematological tests were performed at least once every two to three weeks while receiving S-1 treatment and at least once every three months for five years after treatment finished. For the first five years after surgery, patients received an endoscopy once a year and a CT scan every six months. After relapse, patients underwent a CT scan every two to three months.

### 2.3. Body Composition Analysis

Using the graphic analysis software Ziostation (ZIOSOFT, Tokyo, Japan), we assessed the amount of skeletal muscle and visceral fat on preoperative CT scans. Skeletal muscle mass was assessed at the level of the third lumbar vertebra, whereas visceral fat mass was assessed at the umbilical level. We calculated the visceral adipose tissue index (VAI) and skeletal muscle mass index (SMI) by dividing visceral fat and skeletal muscle mass measured in a single slice by the square of height in m^2^ [1].

Cut-off values for VAI and SMI were determined separately for men and women based on the median values. The cut-off value for VAI was 35.98 cm^2^/m^2^ for men and 28.61 cm^2^/m^2^ for women. Patients whose VAIs were below and above the cut-off value were classified as low-VAIs and high-VAIs, respectively. The cut-off values for SMI were 42.06 cm^2^/m^2^ for men and 33.85 cm^2^/m^2^ for women. Additionally, patients with an SMI below and above the cut-off value were classified as having a low or high SMI, respectively.

### 2.4. Outcomes

The primary outcome was overall survival (OS), with secondary outcomes including cancer-specific survival (CSS), other-cause survival (OCS), disease-free survival (DFS), total number of postoperative complications, severe postoperative complications, infectious complications, and postoperative body weight loss (BWL). The OS is the period between surgery and death. The DFS is the period between surgery and recurrence or death. OCS is the period between surgery and non-cancer death. Postoperative complications of Clavien-Dindo (CD) classification grade 2 or higher were recorded as those occurring within 30 days after surgery. CD grade 3 or higher was considered a severe complication. The BWL rate was measured at 1 month, 6 months, and 1 year postoperatively.

### 2.5. Statistical Analyses

We used PSM to account for differences in patient background and to reduce selection bias. A logistic regression model with the following covariates was used to evaluate the propensity score: sex, age, surgical procedure, surgical approach, clinical stage, comorbidities, lymph node dissection, and SMI; body mass index (BMI) and VAI were excluded. The nearest-neighbor matching method was used, and the two groups were matched one-to-one. The caliper size was 0.20. After matching, we compared the postoperative outcomes between the two groups. The Mann–Whitney U test was used to compare patient characteristics and postoperative outcomes for continuous variables and the chi-square test or Fisher’s exact test for categorical variables. We used the log-rank test for Kaplan–Meier survival analyses. To identify prognostic factors for OS, we used a forward stepwise procedure of Cox proportional hazards regression for multivariate analysis and calculated hazard ratios (HRs). EZR software was used to perform all statistical analyses, which is based on R (The R Foundation for Statistical Computing, Vienna, Austria) and R commander [23]. The statistical significance was set at *p* < 0.05.

## 3. Results

### 3.1. Patient Background

This study’s flowchart is shown in Figure 1. A total of 417 patients met the eligibility criteria, and 209 (50.1%) and 208 (49.9%) were assigned to the low- and high-VAI groups, respectively. Following PSM, both groups had 155 patients. Patient background is shown in Table 1. Before matching, the high-VAI group had a higher BMI (*p* < 0.001), a greater number of patients with diabetes (*p* = 0.018), a higher SMI (*p* < 0.001), and a higher VAI (*p* < 0.001). After matching, there was no significant difference in factors other than BMI and VAI that were not adjusted.

### 3.2. Comparison of Postoperative Outcomes after Matching

The postoperative outcomes after matching are shown in Table 2. There was no difference in pathological findings, operating time, and intraoperative blood loss. A higher number of severe postoperative complications were reported in the high-VAI group (*p* = 0.026). BWL rates were significantly higher in the high-VAI group at 6 months and 1 year (*p* < 0.001 and *p <* 0.001, respectively). There was no difference in the rate of postoperative chemotherapy, but the completion rate for one year was significantly higher in the high-VAI group (*p* = 0.007).

### 3.3. Long-Term Outcomes According to VAI after Matching

The median duration of follow-up was 48 months (interquartile range, 21–60 months). The OS was significantly better in the high-VAI group (HR 0.611, 95%CI 0.403–0.928, *p* = 0.021). There was no difference in the OCS (HR 0.691, 95%CI 0.331–1.440, *p* = 0.323), but the CSS was significantly better for the high-VAI group (HR 0.563, 95%CI 0.339–0.933, *p* = 0.026). The DFS was better in the high-VAI group (HR 0.678, 95%CI 0.470–0.977, *p* = 0.037) (Figure 2).

### 3.4. Long-Term Outcomes According to Severe Complications after Matching

The OS was worse in the severe complications group (HR 2.223, 95%CI 1.295–3.816, *p* = 0.004) (Figure 3a). The OS rates, stratified for both VAI and severe complications, are shown in Figure 3b. Patients with high-VAI without severe complications had the best survival rates, and patients with low-VAI with severe complications had the worst survival rates (*p* = 0.005).

### 3.5. Long-Term Outcomes Stratified by VAI and pStage after Matching

The OS rates, stratified for both VAI and pStage, are shown in Figure 4. In pStage III, patients with high-VAI had better survival rates than patients with low-VAI (*p* < 0.001).

### 3.6. Prognostic Factors for OS

The results of multivariate analysis with forward stepwise procedure of the prognostic factors for OS in all patients without matching are shown in Table 3. Multivariate analysis showed that age ≥ 70 years (HR = 2.101, 95%CI 1.399–3.155, *p* < 0.001), open surgery (HR 2.091, 95%CI 1.411–3.098, *p* < 0.001), pStage ≥ III (HR 4.110, 95%CI 1.411–3.098, *p* < 0.001), postoperative chemotherapy (HR 0.492, 95%CI 0.315–0.767, *p* = 0.002), diabetes (HR 1.580, 95%CI 1.013–2.465, *p* = 0.044), severe postoperative complications (HR 1.791, 95%CI 1.085–2.958, *p* = 0.023), and high-VAI (HR 0.457, 95%CI 0.307–0.680, *p* < 0.001) were independent prognostic factors for OS.

## 4. Discussion

In this study of patients with advanced gastric cancer, a comparison of the two groups after background adjustment using PSM revealed that preoperative high visceral fat was associated not only with an increase in severe complications but also with an increase in infectious complications. However, both OS and DFS were better in the high visceral fat group despite an increase in postoperative complications, which indicates the existence of an obesity paradox in advanced gastric cancer patients. The results of the multivariate analysis support these findings. This is the first report to show that despite the heightened risk of postoperative complications, increased visceral fat retains an advantage in long-term prognosis.

In the comparison of long-term prognosis, OS and CSS were significantly better in the high-VAI group, and DFS was better. The prognosis in the group with severe complications was worse than that in the group without severe complications. In a stratified analysis, high-VAI without severe complications had the best prognosis. High-VAI with severe complications, low-VAI with severe complications, and low-VAI without severe complications had similar prognoses. These results indicate that high-VAI is a favorable prognostic factor, and that although severe complications are a poor prognostic factor, severe complications may counteract their positive impact on prognosis.

Obesity, customarily defined by BMI, has been previously examined in relation to prognosis in gastric cancer with conflicting results. While some studies showed that a high BMI did not affect prognosis [24,25,26], others showed that a BMI over 25 kg/m^2^ was associated with either a poor prognosis [27] or a good prognosis [28]. This discrepancy may be due to the difficulty in determining whether a patient has a high muscle mass or a high visceral fat mass based on BMI alone. Indeed, visceral adiposity has been shown to be more useful than BMI in predicting postoperative complications [2,7,9], and body composition assessment is considered to be more accurate in predicting postoperative outcomes.

Regarding the relationship between visceral fat mass and prognosis, Harada et al. [29] reported in a study of 507 patients, half with esophageal cancer and half with gastric cancer that low visceral fat, measured by preoperative CT, was associated with poor prognosis. This is because low visceral fat mass may be an indicator of undernutrition, whereas high visceral fat mass is a rich source of energy that can be replenished. One of the reasons visceral fat mass is associated with a good prognosis is because of the extensive and persistent postoperative BWL, which is unique to gastrectomy. After gastrectomy, BWL persists for up to 6 months, during which time skeletal muscle mass decreases, mainly during the immediate postoperative period and up to approximately 3 months, and is replaced by adipose tissue thereafter [29,30,31,32,33,34]. The skeletal muscle mass loss is particularly prominent in the first week [31], followed by visceral fat-center changes in body composition, which are understood as metabolic changes that maintain muscle mass. The greater the rate of postoperative weight loss, the worse the compliance with postoperative treatment [35] and the poorer the survival [36,37,38]. Regarding visceral fat mass, Park et al. [39] reported that a large postoperative loss of visceral fat mass was associated with a poor prognosis. These results suggest that visceral fat mass reflects the amount of energy stored in the whole body, and low preoperative fat or large postoperative fat loss indicates nutritional depletion, leading to a poor prognosis.

BWL was significantly greater in the high-VAI group at 6 months and up to 1 year, while there was no difference in the rate up to 1 month postoperatively. This means that there was no difference in the rate of skeletal muscle mass loss in the acute phase, but there was a difference in the time when visceral fat mass decreased. This is thought to reflect the difference in body size before surgery and may be the result of adjusting for the skeletal muscle mass in the PSM to create a model in which only the visceral fat mass was different. Although it has been reported that the greater the postoperative weight loss, the worse the prognosis [36,37,38], the long-term prognosis was better in the high-VAI group, which had a greater weight loss. This indicates that the positive impact of preoperative visceral fat mass on long-term prognosis is greater than the negative impact of postoperative weight loss.

Regarding the choice of method to measure visceral fat mass, while BMI is a convenient method for determining obesity, it is difficult to distinguish between skeletal muscle mass, visceral fat, and subcutaneous fat using BMI alone. A CT scan has the advantage of evaluating body components separately. CT scans are the gold standard for measuring visceral fat mass [40], and recently, there have been an increasing number of reports of body composition assessed by CT. Kobayashi et al. showed that a single slice of visceral fat area measured at the umbilical level correlated strongly with the overall visceral fat mass [41]. Therefore, the measurement of visceral fat mass in a single slice in this study is likely to reflect visceral fat mass accumulation in the whole body. Since the cut-off value for skeletal muscle mass has been examined using height-corrected indexes [42], visceral fat mass was also calculated using height correction in this study. We used the median as the cut-off value for comparison between the two groups, although there is no validated cut-off value for visceral fat in gastric cancer patients.

The limitations of this study include the following: (1) the fact that it was a single-center retrospective cohort study, (2) the fact that there was no postoperative nutritional support, (3) racial differences, and (4) cut-off values for visceral fat. Only patients with insufficient dietary intake received oral nutrition supplementation in this study. It is also necessary to consider racial differences in body size. Asians have a lower BMI and are less likely to be obese than Europeans, which could have influenced the findings. As a result, the findings must be replicated in other racial groups. Furthermore, the VAI cut-off values need to be validated in additional multicenter cohort studies. To the best of our knowledge, this is the first report showing that increased visceral fat has an advantage over risk of postoperative complications in improving long-term prognosis in patients with gastric cancer who exhibit postoperative BWL. Our results suggest that there is a high need for postoperative nutritional support in the low-VAI group, while the high-VAI group requires preoperative nutritional intervention and prehabilitation to prevent postoperative complications. In the future, we would like to investigate whether a support system that includes exercise and nutritional intervention for preoperative weight loss in the high-VAI group and postoperative nutritional support in the low-VAI group will lead to longer DFS and OS.

## 5. Conclusions

Preoperative high visceral fat mass increased postoperative infectious and severe complications, while at the same time improved OS after gastrectomy in patients with advanced gastric cancer.

## Figures and Tables

**Figure 1 nutrients-14-04236-f001:**
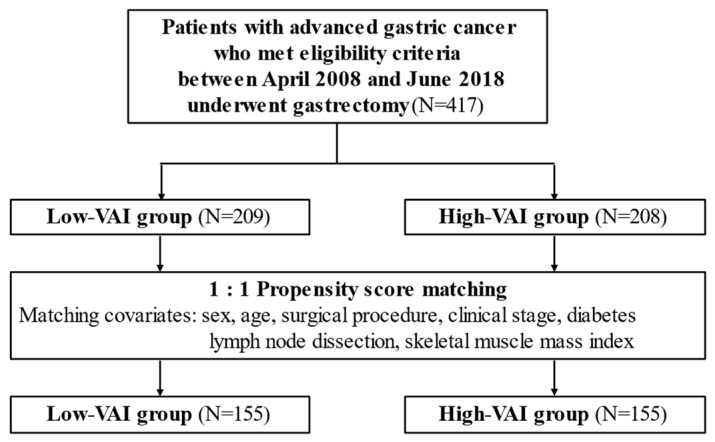
Study design.

**Figure 2 nutrients-14-04236-f002:**
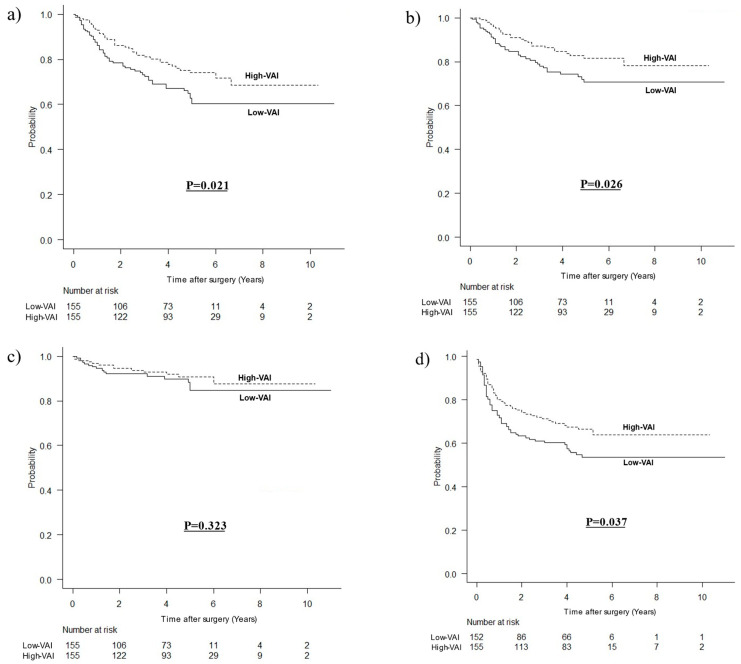
Kaplan–Meier survival curves according to visceral adipose tissue after matching. (**a**) For overall survival, (**b**) for cancer-specific survival, (**c**) for other-cause survival, (**d**) for disease-free survival.

**Figure 3 nutrients-14-04236-f003:**
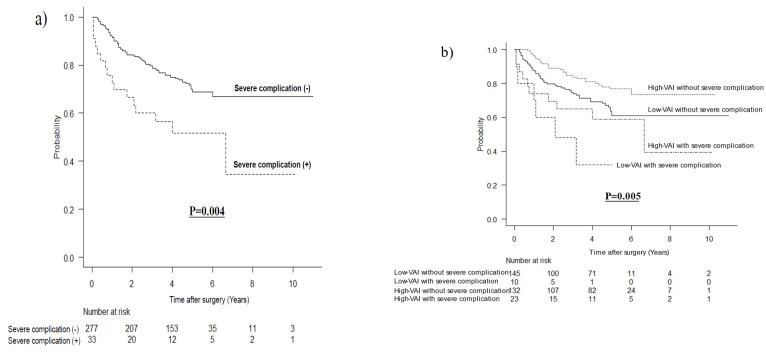
Kaplan-Meier survival curves for overall survival. (**a**) according to severe postoperative complications after matching; (**b**) stratified by visceral adipose tissue and severe postoperative complications after matching.

**Figure 4 nutrients-14-04236-f004:**
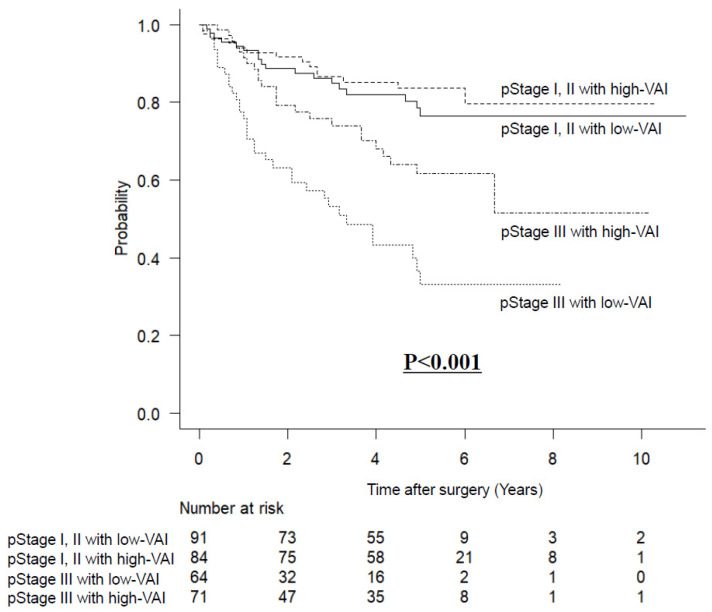
Kaplan–Meier survival curves for overall survival stratified by visceral adipose tissue and pathological stage after matching.

**Table 1 nutrients-14-04236-t001:** Patient characteristics before and after propensity score matching.

	All Patients			After Matching		
	Low-VAI Group (N = 209)	High-VAI Group (N = 208)	*p* Value	Low-VAI Group (N = 155)	High-VAI Group (N = 155)	*p* Value
Sex						
Male Female	140 (67.0%) 69 (33.0%)	140 (67.3%) 68 (32.7%)	1.000	104 (67.1%) 51 (32.9%)	103 (66.5%) 52 (33.5%)	1.000
Age, mean ± SD	67.48 ± 12.09	67.91 ± 9.73	0.689	67.99 ± 11.74	67.63 ± 9.91	0.766
Body mass index, mean ± SD	20.99 ± 2.55	25.06 ± 3.12	<0.001	21.32 ± 2.50	24.62 ± 2.99	<0.001
Surgical approach						
Laparoscopic surgery Open surgery	110 (52.6%) 99 (47.6%)	117 (56.2%) 91 (43.8%)	0.492	87 (56.1%) 68 (43.9%)	84 (54.2%) 71 (45.8%)	0.819
Surgical procedure						
Distal gastrectomy Proximal gastrectomy Total gastrectomy	112 (53.6%) 11 (5.3%) 86 (41.1%)	122 (58.7%) 11 (5.3%) 75 (36.1%)	0.551	85 (54.8%) 7 (4.5%) 63 (40.6%)	87 (56.1%) 9 (5.8%) 59 (38.1%)	0.812
Lymph node dissection						
D1+ D2	82 (39.2%) 127 (60.8%)	101 (48.6%) 107 (51.4%)	0.061	72 (46.5%) 83 (53.5%)	63 (40.6%) 92 (59.4%)	0.359
Clinical stage						
I II III	32 (15.3%) 89 (42.6%) 88 (42.1%)	50 (24.0%) 75 (36.1%) 83 (39.9%)	0.069	29 (18.7%) 62 (40.0%) 64 (41.3%)	32 (20.6%) 52 (33.5%) 71 (45.8%)	0.513
Comorbidity						
CKD	35 (16.7%)	37 (17.8%)	0.797	29 (18.7%)	21 (13.5%)	0.280
COPD	45 (21.5%)	40 (19.2%)	0.627	37 (23.9%)	27 (17.4%)	0.206
Diabetes	30 (14.4%)	49 (23.6%)	0.018	28 (18.1%)	26 (16.8%)	0.881
CHF	8 (3.8%)	13 (6.2%)	0.273	6 (3.9%)	8 (5.2%)	0.786
SMI (cm^2^/m^2^), median (IQR)	38.00 (32.06–43.13)	41.52 (36.73–48.23)	<0.001	39.18 (34.20–44.21)	39.16 (35.46–44.54)	0.572
Low-SMI	127 (60.8%)	81 (38.9%)	<0.001	80 (51.6%)	77 (49.7%)	0.820
VAI (cm^2^/m^2^), median (IQR)	18.18 (8.95–25.10)	51.86 (42.18–66.22)	<0.001	19.69 (9.62–25.84)	50.17 (40.98–60.74)	<0.001

*CHF* chronic heart failure, *CKD* chronic kidney disease, *COPD* chronic obstructive pulmonary disease, *IQR* interquartile range, *SD* standard deviation, *SMI* skeletal muscle mass index, *VAI* visceral adipose tissue index.

**Table 2 nutrients-14-04236-t002:** Comparison of postoperative outcomes after matching.

	Low-VAI Group (N = 155)	High-VAI Group (N = 155)	*p* Value
Pathological stage			
I II III	29 (18.7%) 62 (40.0%) 64 (41.3%)	32 (20.6%) 52 (33.5%) 71 (45.8%)	0.513
Lymph node metastasis			
Absent Present	46 (29.7%) 109 (70.3%)	41 (26.5%) 114 (73.5%)	0.613
Histological type			
Differentiated Undifferentiated	63 (40.6%) 92 (59.4%)	75 (48.4%) 80 (51.6%)	0.209
Operating time (min), median (IQR)	245.0 (207.5–302.5)	245.0 (197.5–325.0)	0.734
Intraoperative blood loss (g), median (IQR)	30.0 (10.0–145.0)	40.0 (17.5–165.0)	0.286
Postoperative complication			
Total number of postoperative complications	28 (18.1%)	41 (26.5%)	0.101
Severe complications	10 (6.5%)	23 (14.8%)	0.026
Infectious complications	17 (11.0%)	29 (18.7%)	0.078
Abdominal abscess	12 (7.7%)	23 (14.8%)	0.072
Incisional surgical site infection	3 (1.9%)	5 (3.2%)	0.723
Anastomotic leakage	6 (3.9%)	9 (5.8%)	0.598
Pancreatic leakage	5 (3.2%)	14 (9.0%)	0.056
Pneumonia	6 (3.9%)	7 (4.5%)	1.000
Ileus	5 (3.2%)	5 (3.2%)	1.000
Cardiovascular complications	1 (0.6%)	2 (1.3%)	1.000
Bleeding complications	1 (0.6%)	5 (3.2%)	0.214
Postoperative chemotherapy Completion rate for one year	99 (63.9%) 80 (76.9%)	99 (63.9%) 94 (91.3%)	1.000 0.007
Postoperative body weight loss (%)			
For 1 month, median (IQR) For 6 months, median (IQR) For 1 year, median (IQR)	7.52 (5.40–11.42) 10.11 (6.09–15.75) 8.87 (5.02–14.77)	8.30 (5.76–11.94) 14.33 (9.50–18.98) 15.79 (10.41–19.62)	0.451 <0.001 <0.001
Pathological stage			
I II III	29 (18.7%) 62 (40.0%) 64 (41.3%)	32 (20.6%) 52 (33.5%) 71 (45.8%)	0.513

*IQR* interquartile range, *VAI* visceral adipose tissue index.

**Table 3 nutrients-14-04236-t003:** Result of multivariate analysis of prognostic factors for overall survival.

Variables		Multivariate Analysis		
		HR	95%CI	*p* Value
Age (years)	<70	1 2.101	1.399–3.155	<0.001
	≥70			
Surgical approach	Laparoscopic surgery	1 2.091	1.411–3.098	<0.001
	Open surgery			
Pathological stage	<III	1 4.110	2.675–6.316	<0.001
	≥III			
Postoperative chemotherapy	Absent	1 0.492	0.315–0.767	0.002
	Present			
Diabetes	Absent	1 1.580	1.013–2.465	0.044
	Present			
Postoperative complications	Absent	1 1.791	1.085–2.958	0.023
	Clavien-Dindo ≥ 3			
SMI (cm^2^/m^2^)	High-SMI	1 1.352	0.915–1.999	0.129
	Low-SMI			
VAI (cm^2^/m^2^)	Low-VAI	1 0.457	0.307–0.680	<0.001
	High-VAI			

*CI* confidence interval, *CRP* C-reactive protein, *HR* hazard ratio, *SMI* skeletal muscle mass index, *VAI* visceral adipose tissue index.

## Data Availability

The datasets generated and/or analyzed during the current study are available upon reasonable request from the corresponding author.

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
