# Peer review of "Preoperative High Visceral Fat Increases Severe Complications but Improves Long-Term Prognosis after Gastrectomy for Patients with Advanced Gastric Cancer: A Propensity Score Matching Analysis"

_nutrients, 2022, doi:10.3390/nu14204236_

Round 1

Reviewer 1 Report

Dear Authors

 This is an interesting study regarding visceral fatty and post-operative complication among patients with advanced gastric cancer from Japan. The followings are my comments. 

#1. The high VAI and low VAI was artificially divided into two groups from a single patient population. There should be a validated cut-off level to confirm the findings from this study. It is interesting to know, if the level of VAI was divided into high/medium/low, is the finding the same?

#2. It is interesting to know the specific types of complications in this study, i.e. cardiopulmonary complication, leakage complication, or pulmonary/ intraabdominal infection.

#3. As the study of BMI on the patient survival, it is interesting to know whether the effect of VAI on patient long term survival with different pathological stage ?

Author Response

#1. The high VAI and low VAI was artificially divided into two groups from a single patient population. There should be a validated cut-off level to confirm the findings from this study. It is interesting to know, if the level of VAI was divided into high/medium/low, is the finding the same?

→Thank you for your important remarks. We used the median as our cut-off values because skewed cut-off values, such as the 10th or 25th percentile, would not reveal the overall trend and would result in smaller sample sizes when using propensity scores. It is very interesting to divide the data into three groups, but there is a drawback that the cut-off value goes from one to two. We believe that the cut-off values need to be validated, so we have added them to the limitation section.

#2. It is interesting to know the specific types of complications in this study, i.e. cardiopulmonary complication, leakage complication, or pulmonary/ intraabdominal infection.

→Thank you for pointing this out. We have added a breakdown of postoperative complications to Table 2: there was a trend toward more intra-abdominal infections in the high-VAI group.

#3. As the study of BMI on the patient survival, it is interesting to know whether the effect of VAI on patient long term survival with different pathological stage ?

→Thank you for your suggestion. We have added a survival curve to Figure 4 showing the relationship between pathological stage and visceral fat mass. In pStage III, patients with high-VAI had better survival rates than patients with low-VAI.

Reviewer 2 Report

I would like to thank the authors for this interesing paper on the effect of VAI in prognosis in patients with advanced gastric cancer. The article is globally well written and the results are clearly presented. The decision to use the median VAI and SMI as cut-off is quite questionable, but I agree that there no validated cut-offs for VAI (but there is plenty of literature on SMI). From a methodological point of view, I think that the authors should better explain some of their choices (level of measurements for VAI and SMI, PSM, the use of other-causes survival, total number of complications, etc..) as more extensively reported in the comments below.

Page 2, line 42-44: though the cited article reported a significantly lower incidence of post-op complications in the group who did pre-op exercise, I wouldn't say that the decrease of complications is significant as the study group consisted only of 18 patients.

Page 2, lines 89-91: does this mean that patients did not receive any gastroscopy within the first five years after surgery?

Page 3, lines 93-97: why were muscle mass and visceral fat mass measured at two different levels? They are usually evaluated at the same level.

Page 3, line 110: please define other-cause survival as it is not a commonly used outcome measure.

Statistical analysis: the authors state that they used 'neoadjuvant chemotherapy' as one of the covariates for their PSM, but they previously stated that patients who received pre-operative treatment were excluded. Also, the covariates described within the paragraph do not match with those found in Figure 1. Finally, please describe why you decided to use PSM without assessing if there was a real unbalance between groups (for example, through evaluation of standardized mean differences).

Results: I am surprised that there are 13.5% of patients in group Low-VAI and 9.9% in the group High-VAI with stage IV disease. Does that imply the intra-operative discovery of unexpected metastases? Since you focus on stage I-III gastric cancer I think that patients with pathological stage IV disease should be excluded.

The number of patients for the two groups is different between the Tables (171) and the Figures (155).

What do you mean by 'total number of post-operative complications'?

When reporting about survival outcomes, the authors refer to Kaplan-Meier curves but they report hazard ratio (derived from a univariate analysis? the HR for OS does not match with the one reported in Table 3). The results of their multivariate analysis should be better described with a Table (even supplementary).

Author Response

#1. Page 2, line 42-44: though the cited article reported a significantly lower incidence of post-op complications in the group who did pre-op exercise, I wouldn't say that the decrease of complications is significant as the study group consisted only of 18 patients.

→Thank you for pointing this out. We have removed the word "Significance".

#2. Page 2, lines 89-91: does this mean that patients did not receive any gastroscopy within the first five years after surgery?

→Thank you for pointing this out. We have corrected it to "For the first five years after surgery".

#3. Page 3, lines 93-97: why were muscle mass and visceral fat mass measured at two different levels? They are usually evaluated at the same level.

→Thank you for pointing this out. Skeletal muscle mass is commonly measured at the L3 level and visceral fat mass is commonly measured at the umbilical level. The following text is included in the Discussion section. “Kobayashi et al. showed that a single slice visceral fat area measured at the umbilical level correlated strongly with the overall visceral fat mass [42].”

#4. Page 3, line 110: please define other-cause survival as it is not a commonly used outcome measure.

→Thank you for your suggestion, I have added the following to Line 115 regarding the definition of OCS. “OCS was defined as the period between surgery and non-cancer death.”

#5. Statistical analysis: the authors state that they used 'neoadjuvant chemotherapy' as one of the covariates for their PSM, but they previously stated that patients who received pre-operative treatment were excluded. Also, the covariates described within the paragraph do not match with those found in Figure 1. Finally, please describe why you decided to use PSM without assessing if there was a real unbalance between groups (for example, through evaluation of standardized mean differences).

→Thank you for pointing this out. We have corrected the description as neoadjuvant chemotherapy was excluded in this study.

#6. Results: I am surprised that there are 13.5% of patients in group Low-VAI and 9.9% in the group High-VAI with stage IV disease. Does that imply the intra-operative discovery of unexpected metastases? Since you focus on stage I-III gastric cancer I think that patients with pathological stage IV disease should be excluded.

→Thank you for pointing this out. We had excluded pStage IV. Due to our mistake, we posted the Table 1 before the revision. We have corrected it.

#7. The number of patients for the two groups is different between the Tables (171) and the Figures (155).

→Thank you for pointing this out. We have corrected it.

#8. What do you mean by 'total number of post-operative complications'?

→Thank you for pointing this out. Postoperative complications are defined below. “Postoperative complications were defined as Clavien-Dindo classification (CD) grade 2 or higher that occurred within 30 days after surgery.”

#8. When reporting about survival outcomes, the authors refer to Kaplan-Meier curves but they report hazard ratio (derived from a univariate analysis? the HR for OS does not match with the one reported in Table 3). The results of their multivariate analysis should be better described with a Table (even supplementary).

→Thank you for pointing this out. In addition to calculating the survival curves and univariate HRs after matching, we also performed a multivariate analysis on all unmatched patients to calculate HRs. The universality of the results is demonstrated by performing both methods. We have added "without matching" to Line 203.

Round 2

Reviewer 1 Report

The authors improved the questions in the 2nd version. I have no more comments.